# A Quantum-Based Beetle Swarm Optimization Algorithm for Numerical Optimization

**Lin Yu [1,*], Jieqi Ren [2] and Jie Zhang [1]**

1   School of Automation, Nanjing University of Science and Technology, Nanjing 210094, China
2   Institute706, The Second Academy, China Aerospace Science & Industry CORP, Beijing 100854, China
*   Correspondence: yulin@njust.edu.cn

**Featured Application: The algorithm proposed in this paper can be widely used in many fields, such as combinatorial optimization, parameter tuning, path planning, etc.**

**Abstract:** The beetle antennae search (BAS) algorithm is an outstanding representative of swarm intelligence algorithms. However, the BAS algorithm still suffers from the deficiency of not being able to handle high-dimensional variables. A quantum-based beetle swarm optimization algorithm (QBSO) is proposed herein to address this deficiency. In order to maintain population diversity and improve the avoidance of falling into local optimal solutions, a novel quantum representation-based position updating strategy is designed. The current best solution is regarded as a linear superposition of two probabilistic states: positive and deceptive. An increase in or reset of the probability of the positive state is performed through a quantum rotation gate to maintain the local and global search ability. Finally, a variable search step strategy is adopted to speed up the ability of the convergence. The QBSO algorithm is verified against several swarm intelligence optimization algorithms, and the results show that the QBSO algorithm still has satisfactory performance at a very small population size.

**Keywords:** BAS algorithm; QEA; swarm intelligent optimization; numerical optimization

## 1. Introduction

Population-based intelligence algorithms have been widely used in many fields because of their simple principle, easy implementation, strong scalability, and high optimization efficiency, such as in UAV path planning [1–3], combinatorial optimization [4,5], and community detection [6,7]. With the increase in the speed of intelligent computing and the development of artificial intelligence, many excellent intelligent algorithms have been proposed such as the seagull optimization algorithm (SOA) [8], artificial bee colony (ABC) algorithm [9], and gray wolf optimization (GWO) algorithm [10]. In addition, there are several intelligent algorithms that were proposed earlier and developed relatively well, such as the particle swarm optimization (PSO) algorithm [11], genetic algorithm (GA) [12], ant colony optimization (ACO) algorithm [13], starling murmuration optimizer (SWO) [14] algorithm, and simulated annealing (SA) algorithm [15].

In 2017, the BAS algorithm was proposed by Jiang [16]. The largest difference between the BAS algorithm and other intelligent algorithms is that the BAS algorithm only needs one beetle to search. Due to the advantages of having a simple principle, fewer parameters, and less calculation, it has been successfully applied to the following optimization fields. Khan et al., proposed an enhanced BAS with zeroing neural networks for solving constrained optimization problems online [17]. Sabahat et al., solved the shortcomings of the low positioning accuracy of sensors in Internet of Things applications using the BAS algorithm [18]. Khan et al., optimized the trajectory of a five-link biped robot based on the longhorn BAS algorithm [19]. Jiang et al., implemented a dynamic attitude configuration of

a wearable wireless body sensor network through a BAS strategy [20]. Khan et al., proposed a strategy based on the BAS algorithm to search for the optimal control parameters of a complex nonlinear system [21].

Although the BAS algorithm exhibits its unique advantages in terms of the calculation amount and principle, the BAS algorithm drastically reduces the optimization performance and even fails to search with high probability when dealing with multidimensional (more than three-dimensional) problems. The reason is that the BAS algorithm is a single search algorithm and, during the search process, the individual can only move towards one extreme point. In multidimensional problems, there is often more than one extreme point, so it is likely to fall into a local extreme point. On the other hand, the step size during the exploration of the beetle decreases exponentially, which means that the beetles may not be able to jump out of local optima. For these reasons, the BAS algorithm is not equipped to handle complex problems with three or more dimensions.

In order to solve the BAS algorithm's defect of not being able to handle high-dimensional problems, a quantum-based beetle swarm optimization algorithm inspired by quantum evolution is proposed in this paper [22]. On the one hand, quantum bits were used to represent the current best solution as a linear superposition of the probability states of "0" and "1" to improve the early exploration capability of the QBSO algorithm. On the other hand, replacing the individual search with a swarm search and a dynamic step adjustment strategy was introduced to improve the exploitation capability of the beetles. Our work has two main contributions:

- We solved the shortcoming of the BAS algorithm in that it cannot handle high-dimensional optimization problems, and the designed QBSO algorithm has an excellent performance in solving 30-dimensional CEC benchmark functions.
- We used quantum representation to deal well with the balance between the population size in terms of the exploratory power and the algorithmic speed, using fewer individuals to represent more information about the population.

The structure of this article is as follows. Section 2 briefly describes the principle of the BAS algorithm, including the implications of the parameters and the procedure of the BAS algorithm. The innovations of the algorithm (i.e., quantum representation (QR) and quantum rotation gate (QRG)) are presented in Section 3. A series of simulation tests are presented in Section 4. The optimization performance of the QBSO algorithm was evaluated by solving four benchmark functions with three comparison algorithms under different populations. Section 5 is the conclusion.

## 2. Related Work

Although the BAS algorithm shows better performance than other swarm intelligence algorithms in dealing with some low-dimensional problems, as mentioned above the performance of the BAS algorithm in high-dimensional variable optimization problems is poor or even largely ineffective. In order to solve this problem, some researchers have conducted related improvement work.

Khan [17] explained the inability of the BAS algorithm to handle high-dimensional optimization problems. It is claimed that the BAS algorithm has a "virtual particle" limitation, which means it computes the objective function three times per iteration. To overcome around this problem, a continuous-time variant of the BAS was proposed in which the "virtual particle" limitation is eliminated. In this algorithm, a delay factor was introduced. It is critical to keep track of the previous states to determine the current states. Furthermore, the parallel processing nature of a zeroing neural network was integrated with BAS to further boost its search for an optimal solution.

Wang [23] combined the population algorithm with a feedback-based step-size strategy, but this ignores the information interaction between individuals and the population, and just blindly expands the population size, which will inevitably increase the calculation. To accelerate the convergence speed and avoid falling into the local optimal solution, adaptive moment estimation was introduced into the algorithm [24]. The algorithm adjusts different

dimensional steps using ADAM update rules, replacing all dimensional steps with the same size. However, the algorithm only performs well on nonconvex problems. Lin [25] added linearly decreasing inertia weights to the decaying process of the beetle step change to guarantee that the late step size is large enough to jump out of the local optimum. However, this also leads to a slow convergence of the algorithm in the later stages.

Zhou [26] combined the BAS algorithm with the solid annealing process from the perspective of algorithm combination. The inability of the BAS algorithm to handle optimization problems in more than three dimensions is eliminated by complementary advantages. It seems that most current researchers are cleverly circumventing the shortcomings of the BAS through the fusion of multi-intelligent optimization algorithms. Shao [27] proposed a beetle swarm algorithm that divides individuals into elite individuals and other individuals. Each elite individual forms a unique clique, and the individuals in the group will move toward the optimal solution under the guidance of the elite individuals. Yu [28] incorporated the BAS algorithm as a search strategy into the gray wolf optimization algorithm to retain the advantages of the BAS algorithm while avoiding high-dimensional divergence. However, this does not essentially solve the deficiency of the BAS. Lv [29] integrated variation and crossover into the population evolution process to improve the global search for better results. In simple terms, it is a fusion of the BAS algorithm and several features of the genetic algorithm. All of the studies above have similarities: using group search to expand the search dimension and solve the shortcoming of one individual's lack of search ability in higher dimensions. However, this operation is contrary to the essence of the BAS algorithm, which is "simple" and "rapid".

Quantum computing is based on quantum bits, which, unlike the 0.1 bits of a computer, can be a linear superposition of two states. Based on the unique superposition, entanglement, and interference properties of quantum computing, quantum-based algorithms in the field of optimization have great potential to maintain population diversity and prevent falling into local optima [30].

Kundra [31] combined the FIRED algorithm with the cuckoo search optimization algorithm to use quantum superposition state to ensure population diversity. Zamani [32] proposed a quantum-based algorithm for bird navigation optimization. It extends the empirical and social learning in the PSO algorithm to short-term and long-term memory. The probability of the algorithm jumping out of the local optimum is improved by quantum mutation and quantum crossover using the 0–1 representation of quantum for the crossover operation, which is cleverly combined with differential evolution. Inspired by the literature [32], Nadimi-Shahraki extended the QANA algorithm to a binary representation for solving the feature selection problem for large medical datasets, showing satisfactory results [33]. Zhou [34] introduced a truncated mean stabilization strategy based on the quantum particle swarm algorithm [35], while using quantum wave functions to locate the global optimal solution. The improved algorithm improves the population diversity and fusion efficiency. Hao [36] designed the Hamiltonian mapping between the problem domain and the quantum, and solved the general locally constrained combinatorial optimization problem based on the quantum tensor network algorithm. Amaro [37] explored the use of causal cones to reduce the number of qubits required on a quantum computer, and introduced a filtering variational quantum eigen-solver to make combinatorial optimization more efficient. Fallahi [38] used quantum solitons instead of a wave function, and combined them with the PSO algorithm to improve the performance of the algorithm. Soloviev [39] proposed a quantum approximate optimization algorithm to solve the problem of Bayesian network structure learning. A study [40] introduced the quantum computer mechanism into the bat algorithm. Incorporating a chaotic cloud mechanism to accelerate the convergence of positive individuals and chaotic perturbation of negative individuals with the aim of increasing population diversity, the algorithm's ability to handle complex optimization problems is verified through comparative experiments.

In summary, it can be concluded that the integration of population-based BAS with quantum theory is a feasible solution.

## 3. Algorithm

### 3.1. Principle of the BAS Algorithm

The BAS algorithm is inspired by the foraging behavior of beetles in nature (see in Figure 1). Beetles have left and right antennae, which can sense the intensity of food odors in the environment. Beetles move toward food according to the difference in the odor's strength as perceived by the left and right antennae. When the intensity of an odor that is perceived by the left antenna is greater than that by the right antenna, the beetle moves toward the left. Otherwise, the beetle moves toward to the right. The smell of food can be regarded as an objective function. The higher the value of the objective function, the closer the beetle is to the food. The BAS algorithm simulates this behavioral characteristic of beetles and carries out an efficient search process.

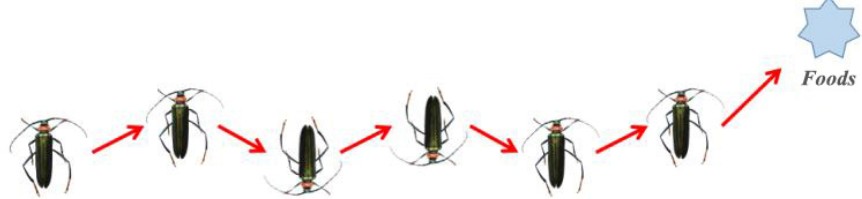

**Figure 1.** Feeding behavior of beetles.

Similar to other intelligent optimization algorithms, the position of an individual beetle in the D-dimensional solution space is $X = (X_1, X_2, \cdots X_D)$. The positions of the left and right antennae of the beetle are defined in the following formula:

$$\begin{cases} X_r = X + l * \vec{d} \\ X_l = X - l * \vec{d} \end{cases} \tag{1}$$

where $l$ denotes the distance between the beetle's center of mass and the antennae; $d$ represents a random unit vector that needs to be normalized to:

$$\vec{d} = \frac{rands(D,1)}{\|rands(D,1)\|_2} \tag{2}$$

Based on the comparison of the intensity of an odor by the left and right antennae, the updated adjustment strategy for the next exploration location of the beetle is as follows:

$$X_{t+1} = X_t + \delta_t * \vec{d} * sign[f(X_r) - f(X_l)] \tag{3}$$

where $t$ represents the current number of iterations of the algorithm; $f(\cdot)$ represents the fitness function; $\delta_t$ is the exploration step at the $t$th iteration; $\varepsilon$ represents the step decay factor, for which the usual value is 0.95; and $sign(\cdot)$ denotes the sign function. The specific definitions of the step and sign function are as follows:

$$\delta_{t+1} = \delta_t \times \varepsilon \tag{4}$$

$$sign(x) = \begin{cases} 1, & if\ x > 0, \\ 0, & if\ x = 0, \\ -1, & otherwise \end{cases} \tag{5}$$

The basic flow of the BAS algorithm is as follows in Figure 2:

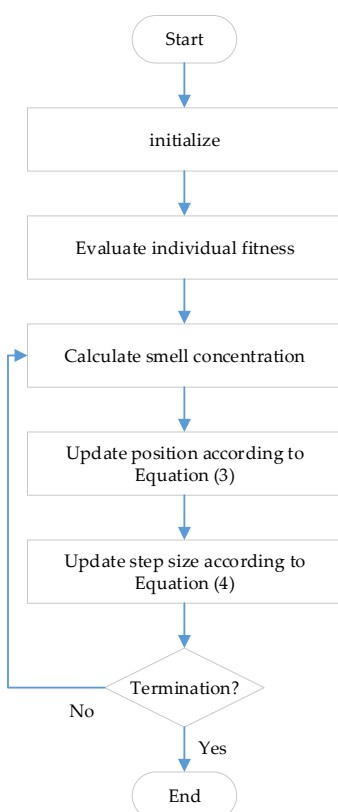

**Figure 2.** BAS algorithm flow chart.

### 3.2. Principle of the QBSO Algorithm

The BAS algorithm is limited by a single individual search and has poor optimization performance in handing multidimensional complex optimization problems. In order to solve this shortcoming, the QBSO algorithm was designed in this study.

### 3.2.1. Quantum Representation

The exploration strategy of the BAS algorithm is similar to other intelligent optimization algorithms, in which balanced exploration and exploitation are achieved by controlling the step size. However, this balancing effect is weak. The premature convergence originates from loss of diversity. Herein, we introduce an alternative approach for preserving diversity of the population. We offer a new comprehension of the concept of optimal solution. The current optimal solution is considered a linear superposition of two probabilistic states: "0" state and "1" state. A qubit of a quantum bit string of length n can be defined as follows:

$$\begin{bmatrix} \alpha_1 & \alpha_2 & \cdots & \alpha_n \\ \beta_2 & \beta_2 & \cdots & \beta_n \end{bmatrix} \tag{6}$$

where $\alpha_i \in [0,1]$, $\beta_i \in [0,1]$, and it satisfies the condition that $\alpha_i^2 + \beta_i^2 = 1 (i = 1, 2, \cdots, n)$; $\alpha^2$ represents the amplitude of the probability in the "1" state; $\beta^2$ represents the amplitude of the probability in the "0" state. The quantum representation of the current global optimal candidate solution can be summarized as follows:

$$x_g^T \triangleq \begin{bmatrix} x_{g,1} & x_{g,2} & \cdots & x_{g,n} \\ \alpha_1 & \alpha_2 & \cdots & \alpha_n \end{bmatrix} \tag{7}$$

To compute the QR observations, a complex function called the wave function $\omega(x, y)$ is introduced here. $|\omega(x, y)|^2$ is the probability density, which represents the probability of a quantum state occurring in the corresponding space and time.

$$|\omega(x_i)|^2 = \frac{1}{\sqrt{2\pi}\sigma_i} \exp\left(-\frac{(x_i - \mu_i)^2}{2\sigma_i}\right), \, i = 1, 2, \cdots, n \tag{8}$$

where $\mu_i$ is the value of the function expectation; $\sigma_i$ represents the standard deviation of the function. The formula for calculating the observed value of the current global optimal solution is as follows:

$$\hat{x}_{g,i} = rand \times |\omega(x_i)|^2 \times (x_{i,\max} - x_{i,\min}) \tag{9}$$

where the expected value of the wave function calculation process can be expressed as $X_{g,i}$ and the variance value as $\sigma_i^2(|\varphi_i\rangle)$.

$$\sigma_i^2(|\psi_i\rangle) = \begin{cases} 1 - |\alpha_i|^2, \, if \, |\psi_i\rangle = |0\rangle, \\ |\alpha_i|^2, \quad if \, |\psi_i\rangle = |1\rangle, \end{cases} \tag{10}$$

The observations of $|\varphi_i\rangle$ using a stochastic process are:

$$|\psi_i\rangle = \begin{cases} |0\rangle, \, \text{if } rand \leq \alpha_i^2 \\ |1\rangle, \, \text{if } rand > \alpha_i^2 \end{cases} \tag{11}$$

The direction of the convergence for each beetle is determined by observing the individuals with the current global optimal solution:

$$d_{j,c} = \hat{x}_{g,i} - x_t \tag{12}$$

$$X_{t+1} = X_t + \delta_t * \vec{d} * \text{sign}[f(X_r) - f(X_l)] + d_{j,c} \tag{13}$$

### 3.2.2. Quantum Rotation Gate

In the quantum genetic algorithm, since the chromosomes under the action of quantum coding are no longer in a single state, the traditional selection, crossover, and mutation operations cannot be continued. Therefore, a QRG is employed to act on the fundamental state of the quantum chromosome to make them interfere with each other and change the phase, thus changing the distribution domain of $\alpha_i$.

Here, QRG is also used to update the probability amplitude of the optimal solution. By increasing the rotation angle, the probability amplitude of $\alpha_i$ is improved. In this way, the convergence rate of individuals toward the global optimal solution is accelerated. At the beginning of the algorithm, the corresponding probability amplitudes of $\alpha_i$ and $\beta_i$ are set to $\sqrt{2}/2$. If the global optimal solution changes after the end of the iteration, $\alpha_i$ is increased by the QRG. Otherwise, all probability amplitudes are reset to the initial value to prevent the algorithm from falling into the local optimum. The update strategy of the QRG is as follows:

$$\alpha_i(t+1) = [\cos(\Delta\theta) - \sin(\Delta\theta)] \left[\frac{\alpha_i(t)}{\sqrt{1 - [\alpha_i(t)]^2}}\right] \tag{14}$$

$$\alpha_i(t+1) = \begin{cases} \sqrt{\eta}, & \text{if } \alpha_i(t+1) < \sqrt{\eta}, \\ \alpha_i(t+1), & \text{if } \sqrt{\eta} \leq \alpha_i(t+1) \leq \sqrt{1-\eta}, \\ \sqrt{1-\eta}, & \text{if } \alpha_i(t+1) > \sqrt{1-\eta}, \end{cases} \tag{15}$$

where $\eta \in [0, 1]$, which is usually a constant; $\Delta\theta$ is the rotation angle of the QRG, which is equivalent to the step size defining the convergence rate toward the current best solution. Briefly, QRA is considered a variation operator here to enhance the probability of obtaining

a positive optimal solution. If successive iterations are still the current optimal solution, $\alpha$ is increased by QRA, while indicating an increase in the probability of the current optimal solution becoming the global optimal solution. Otherwise, $\alpha$ is reset to maintain vigilance against falling into a local optimum.

In addition, the search step size of the BAS algorithm also affects the convergence rate of the algorithm. If the step size is too large, the convergence rate of the QBSO algorithm will be reduced. If the step size is too small, it may lead to search failure. Therefore, this study changed the step size updating strategy: when the global optimal solution changes, the step size is updated according to Formula (16). Otherwise, the decay of the step size accelerates. In order not to affect the search accuracy, the value of $\varepsilon_{min}$ is set to 0.8 according to the original study. The flow of the QBSO algorithm is shown in Figure 3.

$$\delta_{t+1} = \begin{cases} \delta_t \times \varepsilon, & if \ \hat{x}_g \ not \ changed \\ \delta_t \times \varepsilon_{\min}, & if \ \hat{x}_g \ changed \end{cases} \tag{16}$$

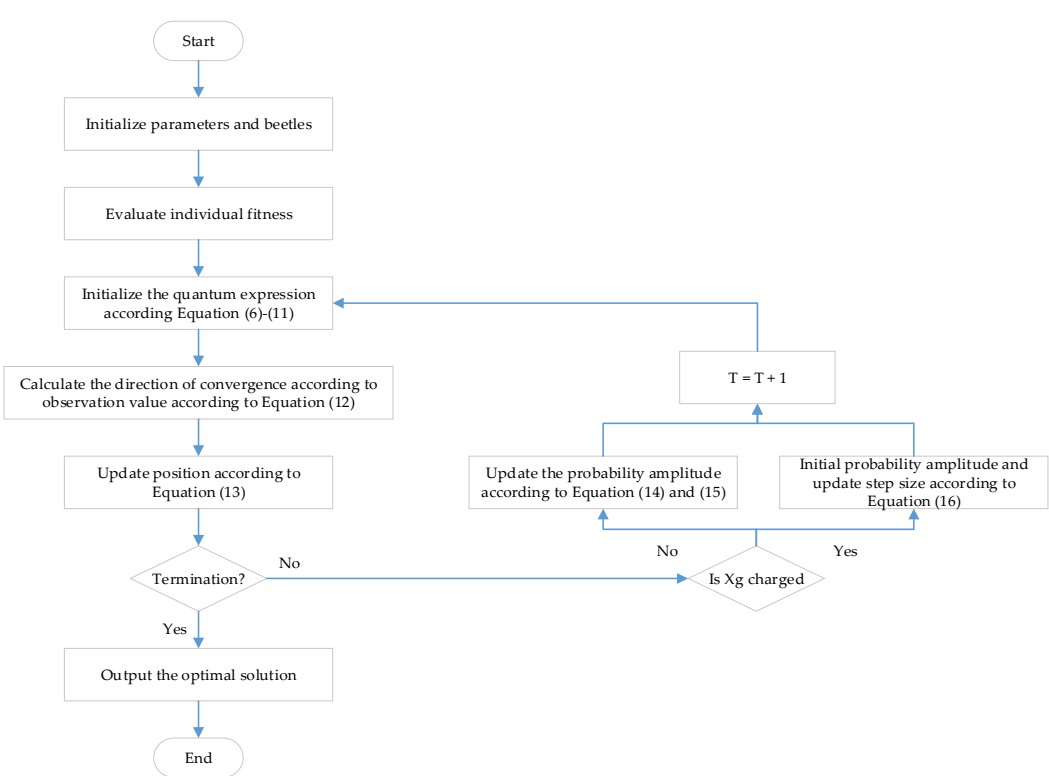

**Figure 3.** Procedure of the QBSO algorithm.

### 3.3. Computational Complexity Analysis

The main time complexity in the QBSO algorithm is within the while loop step. Let n denote the population size and $D$ denote the number of decision variables. The complexity of calculating the direction of the convergence $d_{i,c}$ is $O(D_n)$. The complexity of updating the location information $x$ is $O(n)$. The complexity of the quantum revolving gate is $O(n^2)$. When dealing with large-scale optimization problems, $D \gg n$. According to the operation rules of the symbol $O$, the worst-case time complexity for the QBSO can be simplified as $O(TD)$. When dealing with nonlarge-scale optimization problems, $D \approx n$. The worst-case time complexity for the QBSO algorithm can be simplified as $O(T * n(d + n))$.

## 4. Experiment

Since the BAS algorithm cannot solve high-dimensional complex optimization problems, it cannot be used for simulation comparison experiments with the QBSO algorithm. Therefore, the pigeon-inspired optimization algorithm (PIO), seagull optimization algo-

rithm (SOA), gray wolf optimization algorithm (GWO), and beetle swarm optimization (BSO) algorithm [41] were chosen as the comparison objects. To ensure the validity of the experimental results, the common parameter settings were identical in all algorithms, where the rotation angle in the QBSO was $-11°$ [22]. The other algorithm parameters remained the same as in the original literature. We used trial and error to select the number of iterations. In the context of a population size of 30, the Griewank function was optimized with different numbers of iterations (see Figure 4).

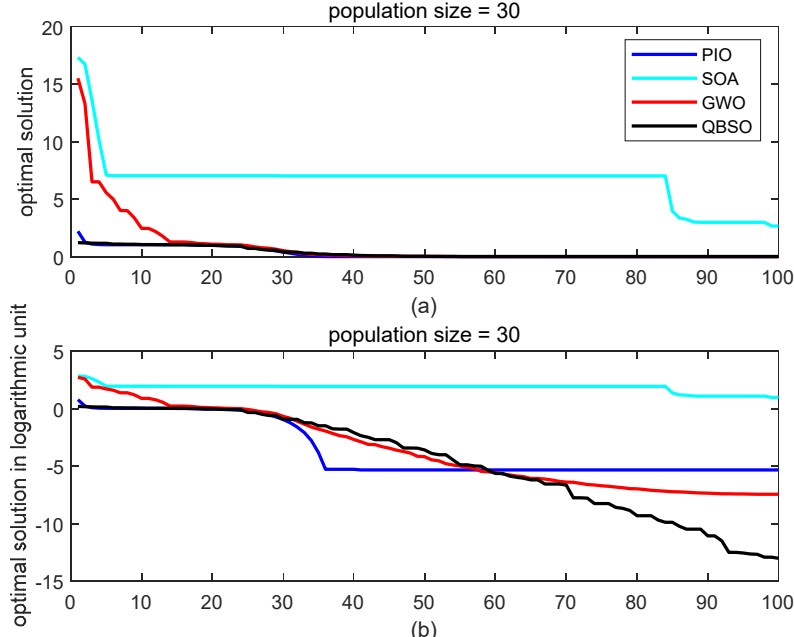

**Figure 4.** Convergence of the optimal solution of the Griewank function at different iterations. (**a**) Optimal solution in natural number unit; (**b**) optimal solution in logarithmic unit.

When the number of iterations is 100, all algorithms basically converge to near the global optimal solution. Each algorithm is comparable. Therefore, we set the iteration number to 100.

To ensure that the PIO, SOA, and GWO algorithms are well explored and developed, researchers usually maintain the population size of the algorithms between 30 and100. If the population is too small, it will affect the searching and convergence abilities of the algorithm. Too large of a population can waste population resources and increase the search time. In order to verify that quantum expression can represent richer population forms with fewer individuals, a comparison experiment with the population size set to 8 and 30 was performed.

We conducted multiple comparison experiments on both the unimodal unconstrained optimization problem and multimodal unconstrained optimization problem. The unimodal benchmark function has only one optimal solution and can be used to detect how quickly the algorithm converges to the vicinity of the optimal solution. The multimodal benchmark function has multiple optimal solutions and is used to detect the ability of the algorithm to jump out of the local optimum.

### 4.1. Unimodal Unconstrained Optimization

The unimodal function only has a single optimum solution and the benchmark problems can be seen in Table 1 [42], where the decision variables of $F_1$ and $F_3$ are 2-dimensional and the other functions are 30-dimensional. The formulation of the functions $f(y)$, their global minima $f(y)_{min}$, and the value of the estimated variables $y(t)$ are shown in Table 1.

**Table 1.** Unimodal benchmark functions.

| Name | Formulation f(y) | $f(y)_{min}$ | y(t) |
|:---:|:---:|:---:|:---:|
| $F_1$ | $-200e^{-0.2\sqrt{y_1^2+y_2^2}}$ | $-200$ | $\{0,0\}$ |
| $F_2$ | $\sum_{i=1}^{n-1}\left(y_i^2\right)^{\left(y_{i+1}^2+1\right)}+\left(y_{i+1}^2\right)^{\left(y_i^2+1\right)}$ | $0$ | $\{0,0,\cdots,0\}$ |
| $F_3$ | $-\dfrac{1+cos\left(12\sqrt{y_1^2+y_2^2}\right)}{\left(0.5\left(y_1^2+y_2^2\right)+2\right)}$ | $-1$ | $\{0,0\}$ |
| $F_4$ | $\sum_{i=1}^{n}|y_i|^{i+1}$ | $0$ | $\{0,0,\cdots,0\}$ |

To demonstrate that the QBSO algorithm can exhibit an excellent optimization performance at a relatively small population size, we conducted comparative experiments with population sizes of 8 and 30 under unimodal optimization problems. Each algorithm was run independently 100 times. The best, worst, average, and variance of the results obtained by each algorithm were collected and used to verify the performance of the algorithm. The optimization results of the unimodal benchmark functions are shown in Tables 2 and 3.

We randomly chose 1 of the 100 independent runs and plotted the algorithm optimization iteration process as a graph, as shown in Figure 4. Considering that when the population size was eight, the optimization results of the PIO algorithm, SOA, and GWO algorithm were so different from the QBSO that it was easy to compress the QBSO into an approximate horizontal line in the figure, we omitted the iterative curve plot here for the population size of eight.

**Table 2.** Results of the unimodal benchmark function experiments (population size = 30).

| Name | Algorithm | Best | Worst | Average | Variance | Time(s) |
|:---:|:---:|:---:|:---:|:---:|:---:|:---:|
| $F_1$ | PIO | $-199.7120$ | $-175.2841$ | $-195.4801$ | $19.4308$ | $0.029$ |
| | SOA | $-199.9893$ | $-45.0261$ | $-185.6132$ | $658.3112$ | $0.010$ |
| | **GWO** | $\mathbf{-200}$ | $\mathbf{-200}$ | $\mathbf{-200}$ | $\mathbf{10^{-28}}$ | $0.011$ |
| | QBSO | $\mathbf{-200}$ | $-199.9999$ | $\mathbf{-200}$ | $10^{-10}$ | $0.087$ |
| | BSO | $\mathbf{-200}$ | $-177.8722$ | $-197.8545$ | $10.3658$ | $0.009$ |
| | BAS | $-199.9965$ | $-199.8666$ | $-199.9398$ | $10^{-4}$ | $0.017$ |
| $F_2$ | PIO | $10^{-4}$ | $31.4922$ | $6.2104$ | $60.2032$ | $0.027$ |
| | SOA | $0.0020$ | $10^4$ | $10^3$ | $10^7$ | $0.018$ |
| | GWO | $0.0017$ | $0.0391$ | $0.0134$ | $10^{-5}$ | $0.026$ |
| | **QBSO** | $\mathbf{10^{-7}}$ | $\mathbf{10^{-5}}$ | $\mathbf{10^{-6}}$ | $\mathbf{10^{-11}}$ | $0.152$ |
| | BSO | $0.1055$ | $1.5872$ | $1.2652$ | $2.8857$ | $0.018$ |
| | BAS | $8.366$ | $33.949$ | $19.747$ | $24.237$ | $0.017$ |
| $F_3$ | PIO | $-0.9998$ | $-0.9291$ | $-0.9509$ | $10^{-4}$ | $0.029$ |
| | SOA | $-1$ | $-0.0352$ | $-0.7232$ | $0.0831$ | $0.010$ |
| | GWO | $-1$ | $-0.9362$ | $-0.9754$ | $10^{-4}$ | $0.011$ |
| | **QBSO** | $\mathbf{-1}$ | $\mathbf{-1}$ | $\mathbf{-1}$ | $\mathbf{10^{-23}}$ | $0.088$ |
| | BSO | $-1$ | $-0.9362$ | $-0.9641$ | $10^{-4}$ | $0.010$ |
| | BAS | $-0.996$ | $-0.465$ | $-0.897$ | $10^{-3}$ | $0.018$ |
| $F_4$ | PIO | $10^{-6}$ | $10^7$ | $10^5$ | $10^{12}$ | $0.066$ |
| | SOA | $0.0227$ | $10^{45}$ | $10^{43}$ | $10^{88}$ | $0.025$ |
| | GWO | $10^{-4}$ | $10^3$ | $27.9445$ | $10^4$ | $0.037$ |
| | **QBSO** | $\mathbf{10^{-19}}$ | $\mathbf{10^{-15}}$ | $\mathbf{10^{-16}}$ | $\mathbf{10^{-31}}$ | $0.181$ |
| | BSO | $10^{-4}$ | $10^4$ | $10^3$ | $10^7$ | $0.025$ |
| | BAS | $19.887$ | $10^4$ | $10^3$ | $10^8$ | $0.017$ |

**Table 3.** Results of the unimodal benchmark function experiments (population size = 8).

| Name | Algorithm | Best | Worst | Average | Variance | Time(s) |
|---|---|---|---|---|---|---|
| | PIO | $-200$ | $-156.4101$ | $-186.9236$ | $126.6304$ | $0.025$ |
| | SOA | $-199.9457$ | $-8.5804$ | $-119.6113$ | $10^3$ | $0.009$ |
| $F_1$ | **GWO** | $\mathbf{-200}$ | $\mathbf{-199.9999}$ | $\mathbf{-200}$ | $\mathbf{10^{-11}}$ | $0.010$ |
| | QBSO | $\mathbf{-200}$ | $-199.9996$ | $\mathbf{-200}$ | $10^{-9}$ | $0.075$ |
| | BSO | $-199.9958$ | $-10^{-4}$ | $-177.3426$ | $10^3$ | $0.004$ |
| | PIO | $0.0082$ | $137.1152$ | $26.5353$ | $771.5456$ | $0.020$ |
| | SOA | $73.2925$ | $10^4$ | $10^4$ | $10^8$ | $0.015$ |
| $F_2$ | GWO | $3.2853$ | $57.7149$ | $16.7590$ | $111.2829$ | $0.015$ |
| | **QBSO** | $\mathbf{10^{-8}}$ | $\mathbf{10^{-6}}$ | $\mathbf{10^{-6}}$ | $\mathbf{10^{-12}}$ | $0.095$ |
| | BSO | $3.4566$ | $67.7782$ | $21.5658$ | $155.9654$ | $0.010$ |
| | PIO | $-1$ | $-0.6185$ | $-0.8981$ | $0.0072$ | $0.025$ |
| | SOA | $-0.9635$ | $-0.0045$ | $-0.2380$ | $0.0722$ | $0.009$ |
| $F_3$ | GWO | $-1$ | $-0.9362$ | $-0.9478$ | $10^{-4}$ | $0.010$ |
| | **QBSO** | $\mathbf{-1}$ | $\mathbf{-1}$ | $\mathbf{-1}$ | $\mathbf{10^{-21}}$ | $0.076$ |
| | BSO | $-1$ | $-0.4877$ | $-0.9238$ | $0.0079$ | $0.003$ |
| | PIO | $10^{-4}$ | $10^{13}$ | $10^{11}$ | $10^{24}$ | $0.031$ |
| | SOA | $10^{10}$ | $10^{48}$ | $10^{47}$ | $10^{95}$ | $0.013$ |
| $F_4$ | GWO | $10^5$ | $10^{19}$ | $10^{17}$ | $10^{36}$ | $0.018$ |
| | **QBSO** | $\mathbf{10^{-20}}$ | $\mathbf{10^{-14}}$ | $\mathbf{10^{-15}}$ | $\mathbf{10^{-30}}$ | $0.115$ |
| | BSO | $10^{17}$ | $10^{49}$ | $10^{47}$ | $10^{97}$ | $0.048$ |

### 4.2. Multimodal Unconstrained Optimization

Multimodal functions contain more than one optimal solution, which will also mean that the algorithm is more likely to fall into a local optimum when optimizing these functions. The population-based intelligence optimization algorithm has an upper hand in optimizing these functions, and this is the idea we improved. Collaborative search among multiple individuals is less likely to fall into local optima than single-individual algorithms such as the BAS algorithm. We dealt with these multimodal benchmark functions with the solution space dimensions set to 30. The formulation of the functions $f(y)$, their global minima $f(y)_{min}$, and the value of the estimated variables y($t$) are shown in Table 4.

**Table 4.** Multimodal benchmark functions.

| Name | Formulation $f(y)$ | $f(y)_{min}$ | y($t$) |
|---|---|---|---|
| Ackley | $-20\exp\left(-0.2\sqrt{\frac{1}{n}\sum\limits_{i=1}^{n}x_i^2}\right)-$ $\exp\left(\frac{1}{n}\sum\limits_{i=1}^{n}\cos(2\pi x_i)\right)+20+\exp(1)$ | $0$ | $\{0,0,\cdots,0\}$ |
| Griewank | $\sum\limits_{i=1}^{n}\frac{x_i^2}{4000}-\prod\limits_{i=1}^{n}cos\left(\frac{x_i}{\sqrt{i}}\right)+1$ | $0$ | $\{0,0,\cdots,0\}$ |
| Rastrigin | $10\text{n}+\sum\limits_{i=1}^{n}\left(y_i^2-10\cos(2\pi y_i)\right)$ | $0$ | $\{0,0,\cdots,0\}$ |
| Quarrtic | $\sum\limits_{i=1}^{n}iy_i^4+random[0,1)$ | $0+\text{rand}$ | $\{\sqrt{i},\sqrt{i},\cdots,\sqrt{i}\}$ |

Each algorithm was run independently 100 times. The optimization results of the multimodal benchmark functions with the population sizes of 30 and 8 are shown in Tables 5 and 6.

**Table 5.** Results of the multimodal benchmark function experiments (population size = 30).

| Name | Algorithm | Best | Worst | Average | Variance | Time(s) |
|------|-----------|------|-------|---------|----------|---------|
| Ackley | PIO | 0.0210 | 5.6406 | 2.4490 | 2.5567 | 0.032 |
| | SOA | 0.0620 | 21.3100 | 19.4798 | 19.0403 | 0.021 |
| | GWO | 20.6624 | 21.1627 | 20.9935 | 0.0081 | 0.030 |
| | **QBSO** | $\mathbf{10^{-4}}$ | **0.0030** | **0.0011** | $\mathbf{10^{-7}}$ | 0.140 |
| | BSO | $10^{-5}$ | 6.1147 | 1.9238 | 2.2163 | 0.023 |
| | BAS | 3.753 | 5.506 | 4.399 | 0.1266 | 0.017 |
| Griewank | PIO | $10^{-4}$ | 0.1750 | 0.0390 | 0.0020 | 0.034 |
| | SOA | $10^{-5}$ | 4.4718 | 1.3344 | 0.8192 | 0.022 |
| | GWO | $10^{-4}$ | 0.1750 | 0.0390 | 0.0020 | 0.029 |
| | **QBSO** | $\mathbf{10^{-9}}$ | $\mathbf{10^{-7}}$ | $\mathbf{10^{-8}}$ | $\mathbf{10^{-15}}$ | 0.149 |
| | BSO | 1.0792 | 5.4872 | 1.6651 | 0.4517 | 0.134 |
| | BAS | 0.371 | 0.949 | 0.655 | 0.0156 | 0.017 |
| Rastrigin | PIO | 5.7557 | 247.5033 | 138.0620 | $10^3$ | 0.037 |
| | SOA | 0.4156 | $10^4$ | $10^3$ | $10^7$ | 0.017 |
| | GWO | 27.9906 | 143.6525 | 55.9303 | 359.9034 | 0.029 |
| | **QBSO** | $\mathbf{10^{-6}}$ | $\mathbf{10^{-4}}$ | $\mathbf{10^{-5}}$ | $\mathbf{10^{-9}}$ | 0.143 |
| | BSO | 5.57 | $10^4$ | $10^3$ | $10^6$ | 0.022 |
| | BAS | 82.358 | $10^2$ | $10^2$ | $10^2$ | 0.017 |
| Quarrtic | PIO | 0.0042 | $10^3$ | $10^5$ | 156.2653 | 0.056 |
| | SOA | 0.0084 | $10^8$ | $10^7$ | $10^{16}$ | 0.030 |
| | GWO | 0.1165 | 1.1138 | 0.3810 | 0.0330 | 0.038 |
| | **QBSO** | $\mathbf{10^{-6}}$ | **0.0027** | $\mathbf{10^{-4}}$ | $\mathbf{10^{-7}}$ | 0.175 |
| | BSO | 0.6687 | $10^8$ | $10^8$ | $10^{16}$ | 0.046 |
| | BAS | $10^2$ | $10^2$ | $10^2$ | $10^4$ | 0.018 |

**Table 6.** Results of the multimodal benchmark function experiments (population size = 8).

| Name | Algorithm | Best | Worst | Average | Variance | Time(s) |
|------|-----------|------|-------|---------|----------|---------|
| Ackley | PIO | 0.0694 | 8.2592 | 4.3592 | 4.2878 | 0.023 |
| | SOA | 11.3070 | 21.3684 | 21.0656 | 1.1158 | 0.016 |
| | GWO | 20.7051 | 21.1816 | 21.0613 | 0.0065 | 0.015 |
| | **QBSO** | $\mathbf{10^{-4}}$ | **0.0015** | $\mathbf{10^{-4}}$ | $\mathbf{10^{-8}}$ | 0.097 |
| | BSO | $10^{-5}$ | 20 | 3.5073 | 25.7410 | 0.006 |
| Griewank | PIO | 0.0011 | 1.0355 | 0.6206 | 0.1467 | 0.029 |
| | SOA | 1.0423 | 13.6958 | 5.2995 | 10.9620 | 0.017 |
| | GWO | 0.1494 | 0.9737 | 0.5470 | 0.0272 | 0.016 |
| | **QBSO** | $\mathbf{10^{-10}}$ | $\mathbf{10^{-8}}$ | $\mathbf{10^{-8}}$ | $\mathbf{10^{-16}}$ | 0.095 |
| | BSO | 1.2868 | 10.8748 | 3.9758 | 2.7785 | 0.007 |
| Rastrigin | PIO | 13.1031 | 312.7558 | 131.6508 | $10^3$ | 0.028 |
| | SOA | 159.2776 | $10^4$ | $10^4$ | $10^8$ | 0.011 |
| | GWO | 85.6812 | 329.1141 | 193.3601 | $10^3$ | 0.016 |
| | **QBSO** | $\mathbf{10^{-6}}$ | $\mathbf{10^{-4}}$ | $\mathbf{10^{-5}}$ | $\mathbf{10^{-9}}$ | 0.093 |
| | BSO | 229.76 | $10^4$ | $10^4$ | $10^7$ | 0.006 |
| Quarrtic | PIO | 0.0047 | $10^4$ | $10^3$ | $10^7$ | 0.026 |
| | SOA | 11.1423 | $10^9$ | $10^8$ | $10^{17}$ | 0.018 |
| | GWO | 62.0423 | $10^4$ | $10^3$ | $10^7$ | 0.018 |
| | **QBSO** | $\mathbf{10^{-5}}$ | **0.0109** | **0.0023** | $\mathbf{10^{-6}}$ | 0.109 |
| | BSO | 10.51 | $10^9$ | $10^8$ | $10^{18}$ | 0.013 |

Similarly, we randomly selected 1 result from the 100 independent runs of the multimodal optimization problem. The iterative process is presented in Figure 5. Here, for the reasons we mentioned above, the iteration curve of the algorithm for a population size of eight is not shown.

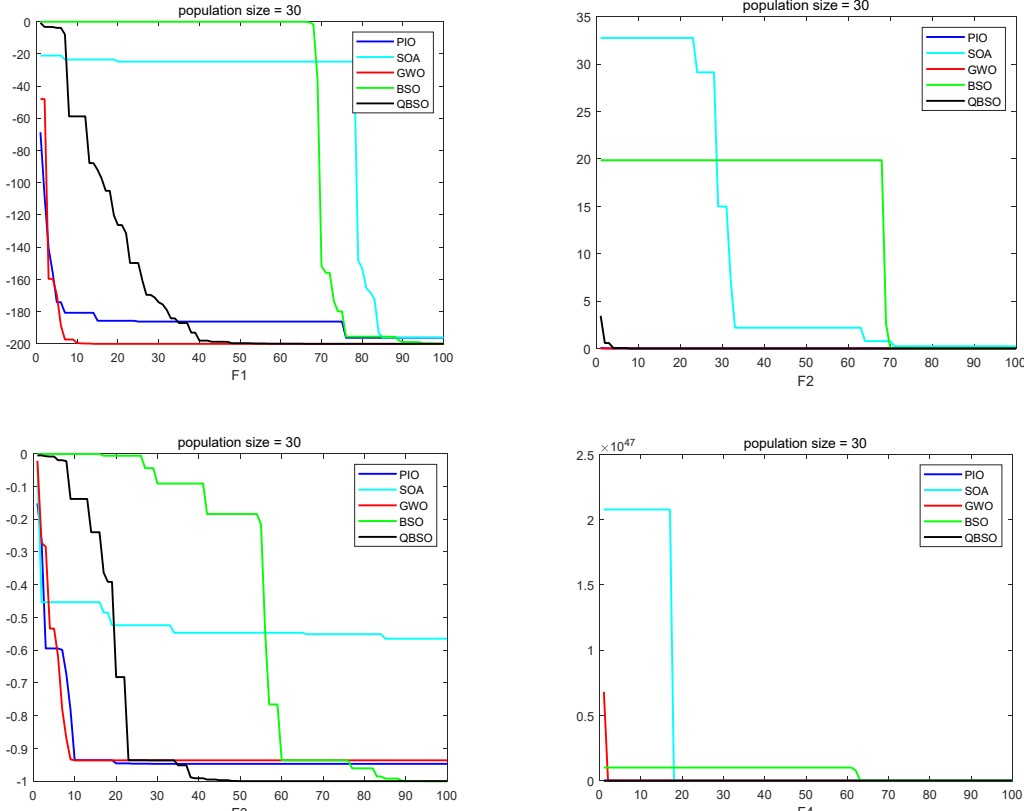

**Figure 5.** The iteration curves when solving unimodal benchmark functions with four algorithms.

For algorithm designers, the accuracy and time consumption of the algorithm are difficult to balance. For population-based optimization algorithms, the larger the dimensionality, the larger the population size that needs to be consumed. It was clear from the computational complexity analysis that the time required to maintain accurate optimization results will grow exponentially. Our goal is to try to trade off dimensionality and time for the decision maker, and to handle the high-dimensional optimization problem with the smallest population size. We conducted a performance comparison of the algorithms in different dimensions with the Rastrigin function, and the results are shown in Table 7.

**Table 7.** Comparison results of the impact of dimensions on algorithm performance.

| D | 5 | 10 | 15 | 20 | 25 | 30 | 35 | 40 | 45 | 50 |
|---|---|---|---|---|---|---|---|---|---|---|
| PIO | 17.5 | 48.1 | 68.7 | 90.1 | $10^2$ | $10^2$ | $10^2$ | $10^2$ | $10^2$ | $10^2$ |
| SOA | 758 | $10^3$ | $10^3$ | $10^4$ | $10^4$ | $10^4$ | $10^4$ | $10^4$ | $10^4$ | $10^4$ |
| GWO | 4.63 | 19.9 | 40.7 | 72.4 | $10^2$ | $10^2$ | $10^2$ | $10^2$ | $10^2$ | $10^2$ |
| BSO | 15.99 | $10^2$ | $10^2$ | $10^2$ | $10^3$ | $10^3$ | $10^3$ | $10^3$ | $10^3$ | $10^3$ |
| QBSO | $10^{-7}$ | $10^{-5}$ | $10^{-5}$ | $10^{-4}$ | $10^{-4}$ | $10^{-4}$ | $10^{-4}$ | $10^{-4}$ | $10^{-3}$ | $10^{-3}$ |

*4.3. Population Diversity Study*

We introduced the population diversity metric to validate the QBSO algorithm diversity metric. The population diversity formula is follows:

$$D_P(t) = \frac{1}{N(t)} \sum_{j=1}^{N(t)} \|x_j(t) - \overline{x}(t)\|_2 \tag{17}$$

where $\overline{x}(t)$ is the mean value of individuals in the current generation. Considering that the BAS is a single individual search algorithm, it cannot constitute a population. Therefore, we chose the BSO algorithm as the diversity comparison algorithm.

## 5. Discussion

It can be observed from Tables 2 and 5 that the QBSO algorithm showed relatively excellent performance in handing both unimodal optimization problems and multimodal optimization problems. This is due to the fact that quantum representation can carry more population information and prevent the loss of diversity. At the same time, the quantum rotation gate as a variational operator can better help the algorithm to jump out of a local optimal solution. The SOA did not perform well because the attack radius of the SOA did not decrease with iteration. This improves the probability of the SOA jumping out of local optimum, but it also loses the fast convergence capability. Therefore, the appropriate iteration size for the QBSO algorithm may not be suitable for the SOA.

As shown by the data in Tables 3 and 6, the PIO algorithm, SOA, GWO algorithm, and BSO algorithm cannot converge to the optimal solution when the population size is eight. On the contrary, the QBSO algorithm continued to perform well.However, there are still several flaws in the QBSO algorithm. From the curves shown in Figures 5 and 6, it can be found that the QBSO algorithm seems to be unable to trade off accuracy and convergence speed. There are two reasons for this: first, the step size adjustment strategy with feedback leads to a slower convergence of the algorithm; second, the variational operation of the quantum rotation gate maintains the diversity but also slightly sacrifices the convergence speed. This will be the focus of our research in the next phase of work.

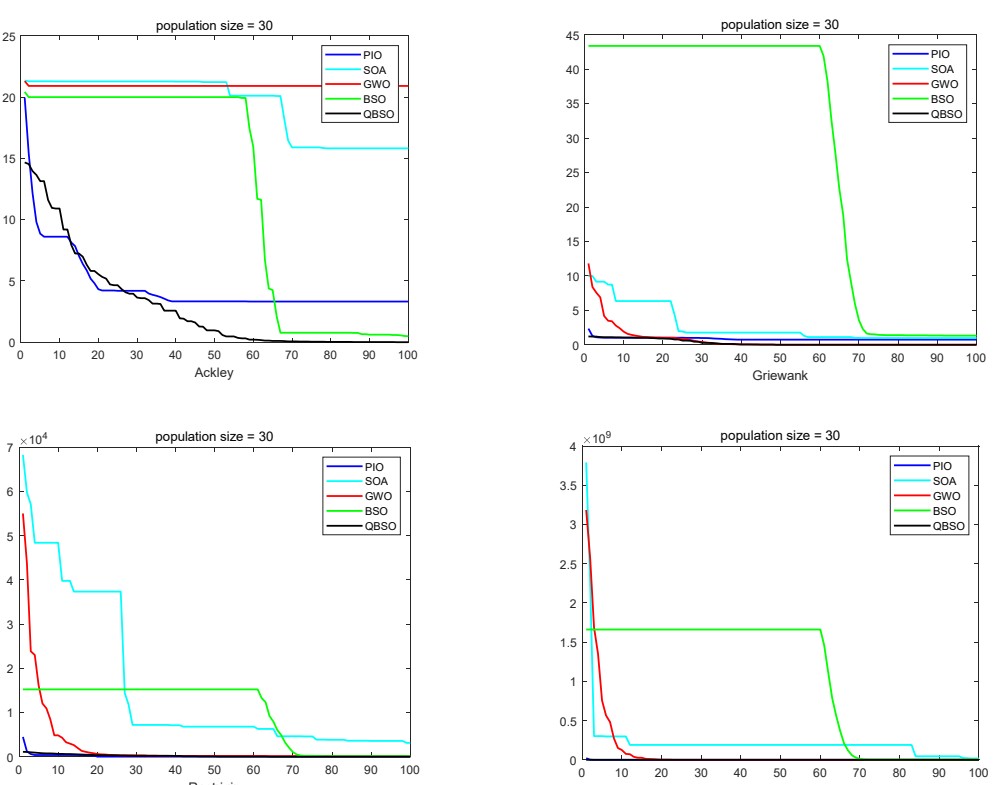

**Figure 6.** The iteration curves when solving the multimodal benchmark functions with four algorithms.

Our design attempted to handle the high-dimensional optimization problem with a minimal population. For further validation, we measured the population diversity and the effect of dimensionality on the performance of the algorithms. Figure 7 shows that the QBSO algorithm had a significant advantage in maintaining population diversity when the number of iterations was less than 40. This was due to the quantum representation of

the QBSO algorithm that enriched the population information and the quantum rotation gate as a variational operator that improved the population variability. Table 7 illustrates that, with increasing dimensionality and unchanged population size, the QBSO algorithm shows the best adaptability, verifying the feasibility of the QBSO for high-dimensional optimization problems.

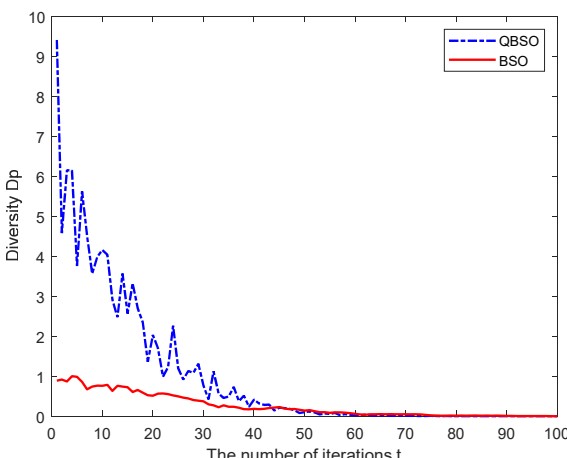

**Figure 7.** Population diversity at different iterations when optimizing the Griewank function with the population size = 30.

## 6. Conclusions

In this paper, we propose the QBSO algorithm to address the inability of the BAS algorithm to handle high-dimensional optimization problems. Quantum representation was introduced into the algorithm, which can carry more population information with small-scale populations. To compare the performance with the PIO, SOA, GWO, and BSO algorithms, multiple comparison experiments with population sizes of 8 and 30 were conducted with unimodal benchmark functions and multimodal benchmark functions as the optimization objectives, respectively. The experimental results show that the QBSO algorithm still had satisfactory optimization capability at a population size of eight. The global convergence ability of the algorithm and the feasibility of the quantum representation were verified. The designed QBSO algorithm can handle high-dimensional optimization problems with low population sizes and still have an excellent optimization performance.

**Author Contributions:** Writing—original draft, L.Y.; revising and critically reviewing for important intellectual content, J.R.; writing—review and editing, J.Z. All authors have read and agreed to the published version of the manuscript.

**Funding:** This research received no external funding.

**Institutional Review Board Statement:** Not applicable.

**Informed Consent Statement:** Not applicable.

**Data Availability Statement:** The datasets are available at: https://github.com/P-N-Suganthan (accessed on 27 January 2021).

**Conflicts of Interest:** The authors declare no conflict of interest.

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
