# Peer review of "A Quantum-Based Beetle Swarm Optimization Algorithm for Numerical Optimization"

_applsci, doi:10.3390/app13053179_

Round 1

Reviewer 1 Report

1. BSA cannot deal with high-dimensional problems, but the author still chooses this algorithm to improve, rather than other optimization algorithms. It is recommended to focus on the advantages of BSA algorithm over other algorithms.

2. The author did not explain why some fixed values were set in QBSO, such as 0.8 in  204 rows, nor did some fixed parameters be selected and analyzed. It is suggested to add.

3. PIO, SOA and GWO algorithms are selected for comparison in the design of comparative experiments, but they are not compared with the improved BSA in other literature, nor with other algorithms after quantum improvement. It is suggested that this part be added.

4. There are too few test functions, so it is recommended to add standard tests.

5. There are not too many interpretations of the test results and no analysis of the statistical images. The source of parameter selection in QBSO has not been explained by the author, which is suggested to be supplemented.

Author Response

Dear Reviewer,

Thank you for your valuable comments for us. We have made changes in accordance with your comments.

Best Wishes,

Authors

Reviewer 2 Report

In this paper, authors proposed QBSO algorithm has excellent performance in solving 30-dimensional CEC benchmark functions. The paper is well structured, however, the following comments should be considered to improve the proposed work.

1- The abbreviations should be explained at the first time of appearance.

2- In line 33, "and" should be written before the word "simulated".

3- In line 84, Khan [22] explain(s).

4- If the proposed algorithm is written in pseudocode, the steps will be much better explained.

5- Authors are encouraged to evaluate the proposed algorithm in terms of one or more engineering problems.

6- The time profile should be tested and compared to other competing algorithms.

7- The complexity analysis of the proposed algorithm should be defined and explained.

Author Response

(The authors gave the same response as above.)

Reviewer 3 Report

The proposed approach to improving the Beetle Antennae Search Algorithm is interesting, promising and will arouse reader interest. However, I suggest the following improvements to the manuscript for better readability and citation.

1. The content of section 3.2.1 Quantum Representation does not allow evaluating the author's contribution to a sufficient extent. I propose to explain directly the author's elements of Quantum Representation. Include in the section references to research that inspired you to use the quantum approach and prompted you to appropriate solutions, if necessary.

2. I suggest supplementing the conclusion with a description of the essence of the way to improve the Beetle Antennae Search Algorithm by proposed Quantum Representation.

3. I propose to present an additional graph of the change in the number of iterations from the size of the population. The study on the example of populations of two sizes, of course, allows us to draw a fundamental conclusion, but does not give a complete picture of the nature of the change in the number of iterations.

4. It is not clear why the authors chose these reference functions for experiments. More detailed justification is required.

5. I propose to check and unify the names of Tables 2, 8, 5 and 6. For example, Table 6 contains the results of experiments with a population size of 8 and not 30. Why is the population size indicated either N or n?

6. The correct full name of the BAS algorithm is Beetle Antennae Search Algorithm (line 11).

7. I offer label by Yes and No directions on flowcharts.

8. Please check the end of the sentence on line 224.

9. The title of section 3.2 must begin with a capital letter.

Author Response

(The authors gave the same response as above.)

Reviewer 4 Report

This study presents a quantum-based beetle swarm optimization algorithm (QBSO) as an improved variant of the beetle antenna search (BAS) algorithm to handle high-dimensional problems. Since this study suffers from a lack of deep experimental analysis, the following comments are suggested.

1.     It is recommended to rewrite the Abstract and mention the novelty and findings of this study.

2.     It is not clear why the new, improved optimizer is needed to solve numerical optimization problems, and it is recommended to state the importance of this study.

3.     The authors claimed the proposed method could establish more diversity throughout the population. It is recommended to provide an experimental evaluation that can support this claim, and a population diversity analysis is recommended.

4.     This study lacks a comprehensive literature review on quantum-based and recent applied optimization algorithms such as qana: quantum-based avian navigation optimizer algorithm, starling murmuration optimizer algorithm wich uses quantum concepts, binary approaches of quantum-based avian navigation optimizer to select effective features from high-dimensional medical data, hybridizing of whale and moth-flame optimization algorithms to solve diverse scales of optimal power flow problem, binary starling murmuration optimizer algorithm to select effective features from medical data, and other similar algorithms.

5.      The performance of the proposed algorithm can be compared with CCSA: Conscious neighborhood-based crow search algorithm for solving global optimization problems.

6.     Equations 13 and 14 are not clear. Please clarify.

7.     The authors claim that the canonical BSA is inadequate for solving high-dimensional problems. But no experimental evaluation of high-dimensional test functions can show the proposed algorithm's superiority in this regard, and large-scale CEC test functions 2013 with dimensions 500 and 1000 are recommended.

8.     The discussion section should be reviewed to clearly show the novelty, findings, and future work.

9.     The proposed algorithm should be compared with the canonical beetle antennae search (BAS).

10.  It is recommended to consider the numbering of equations in Figure 3. For example, Calculate the direction of convergence according to observation value using equation X.

11.  Figure 4 should be checked, and its caption should be revised.

12.  The impact of dimensionality should be investigated in the performance of the proposed algorithm.

Author Response

(The authors gave the same response as above.)

Round 2

Reviewer 1 Report

The modification has been completed as required and the acceptance is agreed.

Author Response

Thank you for your recognition of our work.

Reviewer 2 Report

Thanks for addressing the suggested comments.

Author Response

Dear Reviewer,

Thank you for your comments.

Reviewer 3 Report

The authors provided comprehensive responses to my comments and considered all my suggestions.

I have one minor suggestion to improve the new graph in Figure 4. Important results of the QBSO algorithm are almost invisible due to the unfortunate scale of the y-axis. I suggest using a logarithmic scale for the y-axis in Figure 4.

Author Response

Dear Reviewer,

Thank you for your comments.

We have completed the revision according to your suggestion.

Please see the attachme!Thank You!

Reviewer 4 Report

The authors have responded to the comments partially, and I encourage them to address the following comments sufficiently.

- The literature review can be improved using the recently proposed algorithm. Starling murmuration optimizer algorithm and binary starling murmuration optimizer algorithm to select effective features from medical data use quantum-based movement search strategy, and these algorithms can be considered in the literature review.

-The proposed algorithm should be compared with the canonical beetle antennae search (BAS).

- In the new citation in the revised version, "Nadimi-Shahraki" should be used in the content instead of "Mohammad".

Author Response

(The authors gave the same response as above.)
